# A Discriminative Gaussian Mixture Model with Sparsity

**Hideaki Hayashi & Seiichi Uchida**
Department of Advanced Information Technology
Kyushu University
744, Motooka, Nishi-ku, Fukuoka, 819-0395 JAPAN
{hayashi,uchida}@ait.kyushu-u.ac.jp

## Abstract

In probabilistic classification, a discriminative model based on the softmax function has a potential limitation in that it assumes unimodality for each class in the feature space. The mixture model can address this issue, although it leads to an increase in the number of parameters. We propose a sparse classifier based on a discriminative GMM, referred to as a sparse discriminative Gaussian mixture (SDGM). In the SDGM, a GMM-based discriminative model is trained via sparse Bayesian learning. Using this sparse learning framework, we can simultaneously remove redundant Gaussian components and reduce the number of parameters used in the remaining components during learning; this learning method reduces the model complexity, thereby improving the generalization capability. Furthermore, the SDGM can be embedded into neural networks (NNs), such as convolutional NNs, and can be trained in an end-to-end manner. Experimental results demonstrated that the proposed method outperformed the existing softmax-based discriminative models.

## 1 Introduction

In probabilistic classification, a discriminative model is an approach that assigns a class label $c$ to an input sample $\boldsymbol{x}$ by estimating the posterior probability $P(c \mid \boldsymbol{x})$. The posterior probability $P(c \mid \boldsymbol{x})$ should correctly be modeled because it is not only related to classification accuracy, but also to the confidence of decision making in real-world applications such as medical diagnosis support. In general, the model calculates the class posterior probability using the softmax function after non-linear feature extraction. Classically, a combination of the kernel method and the softmax function has been used. The recent mainstream method is to use a deep neural network for representation learning and softmax for the calculation of the posterior probability.

Such a general procedure for developing a discriminative model potentially contains a limitation due to unimodality. The softmax-based model, such as a fully connected (FC) layer with a softmax function that is often used in deep neural networks (NNs), assumes a unimodal Gaussian distribution for each class (details are shown in Appendix A). Therefore, even if the feature space is transformed into discriminative space via the feature extraction part, $P(c \mid \boldsymbol{x})$ cannot correctly be modeled if the multimodality remains, which leads to a decrease in accuracy.

Mixture models can address this issue. Mixture models are widely used for generative models, with a Gaussian mixture model (GMM) as a typical example. Mixture models are also effective in discriminative models; for example, discriminative GMMs have been applied successfully in various fields, e.g., speech recognition (Tüske et al. 2015; Wang 2007). However, the number of parameters increases if the number of mixture components increases, which may lead to over-fitting and an increase in memory usage; this is useful if we can reduce the number of redundant parameters while maintaining multimodality.

In this paper, we propose a discriminative model with two important properties; multimodality and sparsity. The proposed model is referred to as the sparse discriminative Gaussian mixture (SDGM). In the SDGM, a GMM-based discriminative model is formulated and trained via sparse Bayesian

Figure 1: Comparison of decision boundaries. The black and green circles represent training samples from classes 1 and 2, respectively. The dashed black line indicates the decision boundary between classes 1 and 2 and thus satisfies $P(c = 1 \mid \boldsymbol{x}) = (c = 2 \mid \boldsymbol{x}) = 0.5$. The dashed blue and red lines represent the boundaries between the posterior probabilities of the mixture components.

learning. This learning algorithm reduces memory usage without losing generalization capability by obtaining sparse weights while maintaining the multimodality of the mixture model.

The technical highlight of this study is twofold: One is that the SDGM finds the multimodal structure in the feature space and the other is that redundant Gaussian components are removed owing to sparse learning. Figure 1 shows a comparison of the decision boundaries with other discriminative models. The two-class data are from Ripley's synthetic data (Ripley 2006), where two Gaussian components are used to generate data for each class. The FC layer with the softmax function, which is often used in the last layer of deep NNs, assumes a unimodal Gaussian for each class, resulting in an inappropriate decision boundary. Kernel Bayesian methods, such as the Gaussian process (GP) classifier (Wenzel et al. 2019) and relevance vector machine (RVM) (Tipping 2001), estimate nonlinear decision boundaries using nonlinear kernels, whereas these methods cannot find multimodal structures. Although the discriminative GMM finds multimodal structure, this model retains redundant Gaussian components. However, the proposed SDGM finds a multimodal structure of data while removing redundant components, which leads to an accurate decision boundary.

Furthermore, the SDGM can be embedded into NNs, such as convolutional NNs (CNNs), and trained in an end-to-end manner with an NN. The proposed SDGM is also considered as a mixture, non-linear, and sparse expansion of the logistic regression, and thus the SDGM can be used as the last layer of an NN for classification by replacing it with the fully connected (FC) layer with a softmax activation function.

The contributions of this study are as follows:

- We propose a novel sparse classifier based on a discriminative GMM. The proposed SDGM has both multimodality and sparsity, thereby flexibly estimating the posterior distribution of classes while removing redundant parameters. Moreover, the SDGM automatically determines the number of components by simultaneously removing the redundant components during learning.

- From the perspective of the Bayesian kernel methods, the SDGM is considered as the expansion of the GP and RVM. The SDGM can estimate the posterior probabilities more flexibly than the GP and RVM owing to multimodality. The experimental comparison using benchmark data demonstrated superior performance to the existing Bayesian kernel methods.

- This study connects both fields of probabilistic models and NNs. From the equivalence of a discriminative model based on a Gaussian distribution to an FC layer, we demonstrate that the SDGM can be used as a module of a deep NN. We also demonstrate that the SDGM exhibits superior performance to the FC layer with a softmax function via end-to-end learning with an NN on the image recognition task.

## 2   RELATED WORK AND POSITION OF THIS STUDY

The position of the proposed SDGM among the related methods is summarized in Figure 2. Interestingly, by summarizing the relationships, we can confirm that the three separately developed fields, generative models, discriminative models, and kernel Bayesian methods, are related to each other. Starting from the Gaussian distribution, all the models shown in Figure 2 are connected via

four types of arrows. There is an undeveloped area in the upper right part, and the development of the area is the contribution of this study.

A (unimodal) Gaussian distribution is used as the most naive generative model in machine learning and is the foundation of this relationship diagram. A GMM is the mixture expansion of the Gaussian distributions. Since the GMM can express (almost) arbitrary continuous distributions using multiple Gaussian components, it has been utilized for a long time. Since Gaussian fitting requires numerous parameters, the sparsified versions of Gaussian (Hsieh et al. 2011) and GMM (Gaiffas & Michel 2014) have been proposed.

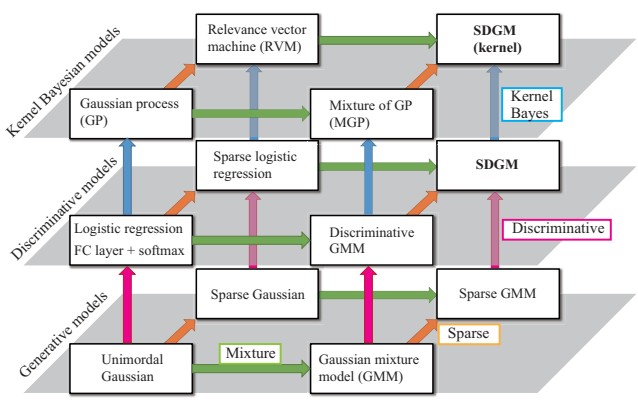

Figure 2: Position of the SDGM among the related methods.

The discriminative models and the generative models are mutually related (Lasserre et al. 2006; Minka 2005). According to Lasserre et al. (2006), the only difference between these models is their statistical parameter constraints. Therefore, given a generative model, we can derive a corresponding discriminative model. For example, discriminative models corresponding to the Gaussian mixture model have been proposed (Axelrod et al. 2006; Bahl et al. 1996; Klautau et al. 2003; Tsai & Chang 2002; Tsuji et al. 1999; Tüske et al. 2015; Wang 2007). They indicate more flexible fitting capability for classification problems than the generative GMM because the discriminative models have a lower statistical bias than the generative models. Furthermore, as shown by Tüske et al. (2015); Variani et al. (2015), these models can be used as the last layer of the NN because these models output the class posterior probability.

From the perspective of the kernel Bayesian methods, the GP classifier (Wenzel et al. 2019) and the mixture of GPs (MGP) (Luo & Sun 2017) are the Bayesian kernelized version of the logistic regression and the discriminative GMM, respectively. The SDGM with kernelization is also regarded as a kernel Bayesian method because the posterior distribution of weights is estimated during learning instead of directly estimating the weights as points, as with the GP and MGP. The RVM (Tipping 2001) is the sparse version of the GP classifier and is the most important related study. The learning algorithm of the SDGM is based on that of the RVM; however, it is extended for the mixture model.

If we use kernelization, the SDGM becomes one of the kernel Bayesian methods and is considered as the mixture expansion of the RVM or sparse expansion of the MGP. Therefore, the classification capability and sparsity are compared with kernel Bayesian methods in Section 4.1. Otherwise, the SDGM is considered as one of the discriminative models and can be embedded in an NN. The comparison with other discriminative models is conducted in Section 4.2 via image classification by combining a CNN.

## 3 SPARSE DISCRIMINATIVE GAUSSIAN MIXTURE (SDGM)

The SDGM takes a continuous variable as its input and outputs the posterior probability of each class, acquiring a sparse structure by removing redundant components via sparse Bayesian learning. Figure 3 shows how the SDGM is trained by removing unnecessary components while maintaining discriminability. In this training, we set the initial number of components to three for each class. As the training progresses, one of the components for each class gradually becomes small and is removed.

### 3.1 NOTATION

Let $x \in \mathbb{R}^D$ be a continuous input variable and $t_c$ ($c \in \{1, \ldots, C\}$, $C$ is the number of classes) be a discrete target variable that is coded in a one-of-$C$ form, where $t_c = 1$ if $x$ belongs to class $c$, $t_c = 0$

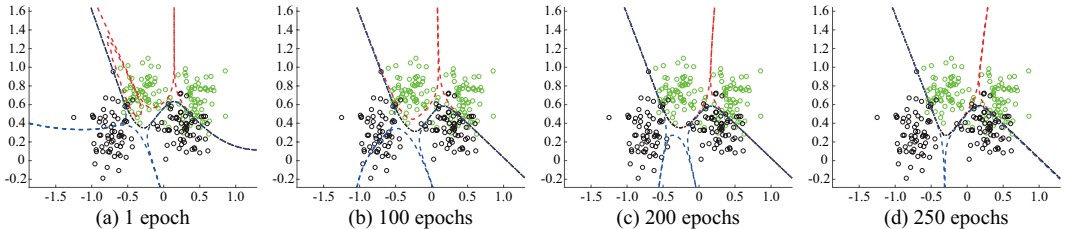

Figure 3: Snapshots of the training process of SDGM. The meanings of lines and circles are the same as Figure 1. There are three components for each class in the initial stage of learning. As the training progresses, one of the components for each class becomes small gradually and is finally removed.

otherwise. Also, let $z_{cm}$ be a discrete latent variable, and $z_{cm} = 1$ when $\boldsymbol{x}$ from class $c$ belongs to the $m$-th component ($m \in \{1, \ldots, M_c\}$, $M_c$ is the number of components for class $c$), $z_{cm} = 0$ otherwise. For simplicity, in this paper, the probabilities for classes and components are described using only $c$ and $m$; e.g., we use $P(c, m \mid \boldsymbol{x})$ instead of $P(t_c = 1, z_{cm} = 1 \mid \boldsymbol{x})$.

## 3.2 Model Formulation

The posterior probabilities of each class $c$ given $\boldsymbol{x}$ are calculated as follows:

$$P(c \mid \boldsymbol{x}) = \sum_{m=1}^{M_c} P(c, m \mid \boldsymbol{x}), \quad P(c, m \mid \boldsymbol{x}) = \frac{\pi_{cm} \exp[\boldsymbol{w}_{cm}^{\mathrm{T}} \boldsymbol{\phi}]}{\sum_{c'=1}^{C} \sum_{m'=1}^{M_{c'}} \pi_{c'm'} \exp[\boldsymbol{w}_{c'm'}^{\mathrm{T}} \boldsymbol{\phi}]}, \tag{1}$$

$$\boldsymbol{\phi} = \left[1, \boldsymbol{x}^{\mathrm{T}}, x_1^2, x_1 x_2, \ldots, x_1 x_D, x_2^2, x_2 x_3, \ldots, x_D^2\right]^{\mathrm{T}}, \tag{2}$$

where $\pi_{cm}$ is the mixture weight that is equivalent to the prior of each component $P(c, m)$. It should be noted that we use $\boldsymbol{w}_{cm} \in \mathbb{R}^H$, which is the weight vector representing the $m$-th Gaussian component of class $c$. The dimension of $\boldsymbol{w}_{cm}$, i.e., $H$, is the same as that of $\boldsymbol{\phi}$; namely, $H = 1 + D(D + 3)/2$.

**Derivation.** Utilizing a Gaussian distribution as a conditional distribution of $\boldsymbol{x}$ given $c$ and $m$, $P(\boldsymbol{x} \mid c, m)$, the posterior probability of $c$ given $\boldsymbol{x}$, $P(c \mid \boldsymbol{x})$, is calculated as follows:

$$P(c \mid \boldsymbol{x}) = \sum_{m=1}^{M_c} \frac{P(c, m) P(\boldsymbol{x} \mid c, m)}{\sum_{c=1}^{C} \sum_{m=1}^{M_c} P(c, m) P(\boldsymbol{x} \mid c, m)}, \tag{3}$$

$$P(\boldsymbol{x} \mid c, m) = \frac{1}{(2\pi)^{\frac{D}{2}} |\boldsymbol{\Sigma}_{cm}|^{\frac{1}{2}}} \exp\left[-\frac{1}{2}(\boldsymbol{x} - \boldsymbol{\mu}_{cm})^{\mathrm{T}} \boldsymbol{\Sigma}_{cm}^{-1} (\boldsymbol{x} - \boldsymbol{\mu}_{cm})\right], \tag{4}$$

where $\boldsymbol{\mu}_{cm} \in \mathbb{R}^D$ and $\boldsymbol{\Sigma}_{cm} \in \mathbb{R}^{D \times D}$ are the mean vector and the covariance matrix for component $m$ in class $c$. Since the calculation inside an exponential function in (4) is quadratic form, the conditional distributions can be transformed as follows:

$$P(\boldsymbol{x} \mid c, m) = \exp[\boldsymbol{w}_{cm}^{\mathrm{T}} \boldsymbol{\phi}], \tag{5}$$

where

$$\boldsymbol{w}_{cm} = \left[ -\frac{D}{2} \ln 2\pi - \frac{1}{2} \ln |\boldsymbol{\Sigma}_{cm}| - \frac{1}{2} \sum_{i=1}^{D} \sum_{j=1}^{D} s_{cmij} \mu_{cmi} \mu_{cmj}, \sum_{i=1}^{D} s_{cmi1} \mu_{cmi}, \right.$$

$$\left. \ldots, \sum_{i=1}^{D} s_{cmiD} \mu_{cmi}, -\frac{1}{2} s_{cm11}, -s_{cm12}, \ldots, -s_{cm1D}, -\frac{1}{2} s_{cm22}, \ldots, -\frac{1}{2} s_{cmDD} \right]^{\mathrm{T}}. \tag{6}$$

Here, $s_{cmij}$ is the $(i, j)$-th element of $\boldsymbol{\Sigma}_{cm}^{-1}$.

## 3.3 Dual Form via Kernelization

Since $\boldsymbol{\phi}$ is a second-order polynomial form, we can derive the dual form of the SDGM using polynomial kernels. By kernelization, we can treat the SDGM as the kernel Bayesian method as described in Section 2.

Let $\boldsymbol{\psi}_{cm} \in \mathbb{R}^N$ be a novel weight vector for the $c, m$-th component. Using $\boldsymbol{\psi}_{cm}$ and the training dataset $\{\boldsymbol{x}_n\}_{n=1}^N$, the weight of the original form $\boldsymbol{w}_{cm}$ is represented as

$$\boldsymbol{w}_{cm} = [\boldsymbol{\phi}(\boldsymbol{x}_1), \cdots, \boldsymbol{\phi}(\boldsymbol{x}_N)]\boldsymbol{\psi}_{cm}, \tag{7}$$

where $\boldsymbol{\phi}(\boldsymbol{x}_n)$ is the transformation $\boldsymbol{x}_n$ of using (2). Then, (5) is reformulated as follows:

$$\begin{aligned} P(\boldsymbol{x} \mid c, m) &= \exp[\boldsymbol{w}_{cm}^{\mathrm{T}}\boldsymbol{\phi}] \\ &= \exp[\boldsymbol{\psi}_{cm}^{\mathrm{T}}[\boldsymbol{\phi}(\boldsymbol{x}_1)^{\mathrm{T}}\boldsymbol{\phi}(\boldsymbol{x}), \cdots, \boldsymbol{\phi}(\boldsymbol{x}_N)^{\mathrm{T}}\boldsymbol{\phi}(\boldsymbol{x})]^{\mathrm{T}}] \\ &= \exp[\boldsymbol{\psi}_{cm}^{\mathrm{T}}K(\mathbf{X}, \boldsymbol{x})], \end{aligned} \tag{8}$$

where $K(\mathbf{X}, \boldsymbol{x})$ is an $N$-dimensional vector that contains kernel functions defined as $k(\boldsymbol{x}_n, \boldsymbol{x}) = \boldsymbol{\phi}(\boldsymbol{x}_n)^{\mathrm{T}}\boldsymbol{\phi}(\boldsymbol{x}) = (\boldsymbol{x}_n^{\mathrm{T}}\boldsymbol{x} + 1)^2$ for its elements and $\mathbf{X}$ is a data matrix that has $\boldsymbol{x}_n^{\mathrm{T}}$ in the $n$-th row. Whereas the computational complexity of the original form in Section 3.2 increases in the order of the square of the input dimension $D$, the dimensionality of this dual form is proportional to $N$. When we use this dual form, we use $N$ and $k(\boldsymbol{x}_n, \cdot)$ instead of $H$ and $\boldsymbol{\phi}(\cdot)$, respectively.

## 3.4 LEARNING ALGORITHM

A set of training data and target value $\{\boldsymbol{x}_n, t_{nc}\}$ $(n = 1, \cdots, N)$ is given. We also define $\boldsymbol{\pi}$ and $\boldsymbol{z}$ as vectors that comprise $\pi_{cm}$ and $z_{ncm}$ as their elements, respectively. As the prior distribution of the weight $w_{cmh}$, we employ a Gaussian distribution with a mean of zero. Using a different precision parameter (inverse of the variance) $\alpha_{cmh}$ for each weight $w_{cmh}$, the joint probability of all the weights is represented as follows:

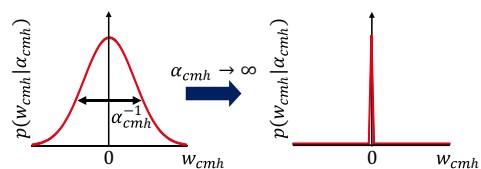

$$\begin{aligned} &P(\boldsymbol{w} \mid \boldsymbol{\alpha}) \\ &= \prod_{c=1}^{C}\prod_{m=1}^{M_c}\prod_{h=1}^{H}\sqrt{\frac{\alpha_{cmh}}{2\pi}}\exp\left[-\frac{1}{2}w_{cmh}{}^2\alpha_{cmh}\right], \end{aligned} \tag{9}$$

Figure 4: Prior for each weight $w_{cmh}$. By maximizing the evidence term of the posterior of $\boldsymbol{w}$, the precision of the prior $\alpha_{cmh}$ achieves infinity if the corresponding weight $w_{cmh}$ is redundant.

where $\boldsymbol{w}$ and $\boldsymbol{\alpha}$ are vectors with $w_{cmh}$ and $\alpha_{cmh}$ as their elements, respectively. During learning, we update not only $\boldsymbol{w}$ but also $\boldsymbol{\alpha}$. If $\alpha_{cmh} \to \infty$, the prior (9) is 0; hence a sparse solution is obtained by optimizing $\boldsymbol{\alpha}$ as shown in Figure 4.

Using these variables, the expectation of the log-likelihood function over $\boldsymbol{z}$, $J$, is defined as follows:

$$J = \mathbb{E}_{\boldsymbol{z}}\left[\ln P(\mathbf{T}, \boldsymbol{z} \mid \mathbf{X}, \boldsymbol{w}, \boldsymbol{\pi}, \boldsymbol{\alpha})\right] = \sum_{n=1}^{N}\sum_{c=1}^{C}r_{ncm}t_{nc}\ln P(c, m \mid \boldsymbol{x}_n),$$

where $\mathbf{T}$ is a matrix with $t_{nc}$ as its element. The variable $r_{ncm}$ in the right-hand side corresponds to $P(m \mid c, \boldsymbol{x}_n)$ and can be calculated as $r_{ncm} = P(c, m \mid \boldsymbol{x}_n)/P(c \mid \boldsymbol{x}_n)$.

The posterior probability of the weight vector $\boldsymbol{w}$ is described as follows:

$$P(\boldsymbol{w} \mid \mathbf{T}, \boldsymbol{z}, \mathbf{X}, \boldsymbol{\pi}, \boldsymbol{\alpha}) = \frac{P(\mathbf{T}, \boldsymbol{z} \mid \mathbf{X}, \boldsymbol{w}, \boldsymbol{\pi}, \boldsymbol{\alpha})P(\boldsymbol{w} \mid \boldsymbol{\alpha})}{P(\mathbf{T}, \boldsymbol{z} \mid \mathbf{X}, \boldsymbol{\alpha})} \tag{10}$$

An optimal $\boldsymbol{w}$ is obtained as the point where (10) is maximized. The denominator of the right-hand side in (10) is called the evidence term, and we maximize it with respect to $\boldsymbol{\alpha}$. However, this maximization problem cannot be solved analytically; therefore we introduce the Laplace approximation described as the following procedure.

With $\boldsymbol{\alpha}$ fixed, we obtain the mode of the posterior distribution of $\boldsymbol{w}$. The solution is given by the point where the following equation is maximized:

$$\begin{aligned} \mathbb{E}_{\boldsymbol{z}}\left[\ln P(\boldsymbol{w} \mid \mathbf{T}, \boldsymbol{z}, \mathbf{X}, \boldsymbol{\pi}, \boldsymbol{\alpha})\right] &= \mathbb{E}_{\boldsymbol{z}}\left[\ln P(\mathbf{T}, \boldsymbol{z} \mid \mathbf{X}, \boldsymbol{w}, \boldsymbol{\pi}, \boldsymbol{\alpha})\right] + \ln P(\boldsymbol{w} \mid \boldsymbol{\alpha}) + \text{const.} \\ &= J - \boldsymbol{w}^{\mathrm{T}}\mathbf{A}\boldsymbol{w} + \text{const.}, \end{aligned} \tag{11}$$

where $\mathbf{A} = \text{diag}\,\alpha_{cmh}$. We obtain the mode of (11) via Newton's method. The gradient and Hessian required for this estimation can be calculated as follows:

$$\nabla\mathbb{E}_{\boldsymbol{z}}\left[\ln P(\boldsymbol{w} \mid \mathbf{T}, \boldsymbol{z}, \mathbf{X}, \boldsymbol{\pi}, \boldsymbol{\alpha})\right] = \nabla J - \mathbf{A}\boldsymbol{w}, \tag{12}$$

---

**Algorithm 1:** Learning algorithm of the SDGM

---

**Input:** Training data set $\mathbf{X}$ and teacher vector $\mathbf{T}$.
**Output:** Trained weight $\boldsymbol{w}$ obtained by maximizing (11).
Initialize the weights $\boldsymbol{w}$, hyperparameters $\boldsymbol{\alpha}$, mixture coefficients $\boldsymbol{\pi}$, and posterior probabilities $\boldsymbol{r}$;
**while** $\boldsymbol{\alpha}$ *have not converged* **do**
    Calculate $J$ using (10);
    **while** $\boldsymbol{r}$ *have not converged* **do**
        **while** $\boldsymbol{w}$ *have not converged* **do**
            Calculate gradients using (12);
            Calculate Hessian (13);
            Maximize (11) w.r.t. $\boldsymbol{w}$;
            Calculate $P(c, m \mid \boldsymbol{x}_n)$ and $P(c \mid \boldsymbol{x}_n)$;
        **end**
        $r_{ncm} = P(c, m \mid \boldsymbol{x}_n)/P(c \mid \boldsymbol{x}_n)$;
    **end**
    Calculate $\Lambda$ using (16);
    Update $\boldsymbol{\alpha}$ using (17);
    Update $\boldsymbol{\pi}$ using (18);
**end**

---

$$\nabla\nabla\mathbb{E}_{\boldsymbol{z}}\left[\ln P(\boldsymbol{w} \mid \mathbf{T}, \boldsymbol{z}, \mathbf{X}, \boldsymbol{\pi}, \boldsymbol{\alpha})\right] = \nabla\nabla J - \mathbf{A}. \tag{13}$$

Each element of $\nabla J$ and $\nabla\nabla J$ is calculated as follows:

$$\frac{\partial J}{\partial w_{cmh}} = (r_{ncm}t_{nc} - P(c, m \mid \boldsymbol{x}_n))\phi_h, \tag{14}$$

$$\frac{\partial^2 J}{\partial w_{cmh}\partial w_{c'm'h'}} = P(c', m' \mid \boldsymbol{x}_n)(P(c, m \mid \boldsymbol{x}_n) - \delta_{cc'mm'})\phi_h\phi_{h'}, \tag{15}$$

where $\delta_{cc'mm'}$ is a variable that takes 1 if both $c = c'$ and $m = m'$, 0 otherwise. Hence, the posterior distribution of $\boldsymbol{w}$ can be approximated by a Gaussian distribution with a mean of $\hat{\boldsymbol{w}}$ and a covariance matrix of $\Lambda$, where

$$\Lambda = -(\nabla\nabla\mathbb{E}_{\boldsymbol{z}}\left[\ln P(\hat{\boldsymbol{w}} \mid \mathbf{T}, \boldsymbol{z}, \mathbf{X}, \boldsymbol{\pi}, \boldsymbol{\alpha})\right])^{-1}. \tag{16}$$

Since the evidence term can be represented using the normalization term of this Gaussian distribution, we obtain the following updating rule by calculating its derivative with respect to $\alpha_{cmh}$.

$$\alpha_{cmh} \leftarrow \frac{1 - \alpha_{cmh}\lambda_{cmh}}{\hat{w}_{cmh}^2}, \tag{17}$$

where $\lambda_{cmh}$ is the diagonal component of $\Lambda$. The mixture weight $\pi_{cm}$ can be estimated using $r_{ncm}$ as follows:

$$\pi_{cm} = \frac{1}{N_c}\sum_{n=1}^{N_c} r_{ncm}, \tag{18}$$

where $N_c$ is the number of training samples belonging to class $c$. As described above, we obtain a sparse solution by alternately repeating the update of hyper-parameters, as in (17) and (18), and the posterior distribution estimation of $\boldsymbol{w}$ using the Laplace approximation. As a result of the optimization, some of $\alpha_{cmh}$ approach to infinite values, and $w_{cmh}$ corresponding to $\alpha_{cmh}$ have prior distributions with mean and variance both zero as shown in (4); hence such $w_{cmh}$ are removed because their posterior distributions are also with mean and variance both zero. During the procedure, the $\{c, m\}$-th component is eliminated if $\pi_{cm}$ becomes 0 or all the weights $w_{cmh}$ corresponding to the component become 0.

The learning algorithm of the SDGM is summarized in Algorithm 1. In this algorithm, the optimal weight is obtained as maximum a posterior solution. We can obtain a sparse solution by optimizing the prior distribution set to each weight simultaneously with weight optimization.

## 4 EXPERIMENTS

### 4.1 COMPARATIVE STUDY USING BENCHMARK DATA

To evaluate the capability of the SDGM quantitatively, we conducted a classification experiment using benchmark datasets. The datasets used in this experiment were Ripley's synthetic data (Ripley

Table 1: Recognition error rate (%) and number of nonzero weights

| | Accuracy (error rate (%)) | | | | Sparsity (number of nonzero weights) | | | |
| | | Baselines | | | | | Baselines | | |
| Dataset | SDGM | GP | MGP | RVM | SDGM | GP | MGP | RVM |
|---|---|---|---|---|---|---|---|---|
| Ripley | **9.1** | **9.1** | 9.3 | 9.3 | 6 | 250 | 1250 | **4** |
| Waveform | 10.1 | 10.4 | **9.6** | 10.9 | **11.0** | 400 | 2000 | 14.6 |
| Banana | 10.6 | **10.5** | 10.7 | 10.8 | **11.1** | 400 | 2000 | 11.4 |
| Titanic | **22.6** | **22.6** | 22.8 | 23.0 | 74.5 | 150 | 750 | **65.3** |
| Breast Cancer | **29.4** | 30.4 | 30.7 | 29.9 | 15.7 | 200 | 1000 | **6.3** |
| Normalized mean | **1.00** | 1.01 | 1.01 | 1.03 | 1.00 | 25.76 | 128.79 | **0.86** |

2006) (Ripley hereinafter) and four datasets cited from (Rätsch et al. 2001); Banana, Waveform, Titanic, and Breast Cancer. Ripley is a synthetic dataset that is generated from a two-dimensional ($D = 2$) Gaussian mixture model, and 250 and 1,000 samples are provided for training and test, respectively. The number of classes is two ($C = 2$), and each class comprises two components. The remaining four datasets are all two-class ($C = 2$) datasets, which comprise different data sizes and dimensionality. Since they contain 100 training/test splits, we repeated experiments 100 times and then calculated average statistics.

For comparison, we used three kernel Bayesian methods: a GP classifier, an MPG classifier (Tresp 2001; Luo & Sun 2017), and an RVM (Tipping 2001), which are closely related to the SDGM from the perspective of sparsity, multimodality, and Bayesian learning, as described in Section 2. In the evaluation, we compared the recognition error rates for discriminability and the number of nonzero weights for sparsity on the test data. The results of RVM were cited from (Tipping 2001). By way of summary, the statistics were normalized by those of the SDGM, and the overall mean was shown.

Table 1 shows the recognition error rates and the number of nonzero weights for each method. The results in Table 1 demonstrated that the SDGM achieved a better accuracy on average compared to the other kernel Bayesian methods. The SDGM is developed based on a Gaussian mixture model and is particularly effective for data where a Gaussian distribution can be assumed, such as the Ripley dataset. Since the SDGM explicitly models multimodality, it could more accurately represent the sharp changes in decision boundaries near the border of components compared to the RVM, as shown in Figure 1. Although the SDGM did not necessarily outperform the other methods in all datasets, it achieved the best accuracy on average. In terms of sparsity, the number of initial weights for the SDGM is the same as MGP, and the SDGM reduced 90.0–99.5% of weights from the initial state due to the sparse Bayesian learning, which leads to drastically efficient use of memory compared to non-sparse classifiers (GP and MGP). The results above indicated that the SDGM demonstrated generalization capability and a sparsity simultaneously.

## 4.2 IMAGE CLASSIFICATION

In this experiment, the SDGM is embedded into a deep neural network. Since the SDGM is differentiable with respect to the weights, SDGM can be embedded into a deep NN as a module and is trained in an end-to-end manner. In particular, the SDGM plays the same role as the softmax function since the SDGM calculates the posterior probability of each class given an input vector. We can show that a fully connected layer with the softmax is equivalent to the discriminative model based on a single Gaussian distribution for each class by applying a simple transformation (see Appendix A), whereas the SDGM is based on the Gaussian mixture model.

To verify the difference between them, we conducted image classification experiments. Using a CNN with a softmax function as a baseline, we evaluated the capability of SDGM by replacing softmax with the SDGM. We also used a CNN with a softmax function trained with $L_1$ regularization, a CNN with a large margin softmax (Liu et al. 2016), and a CNN with the discriminative GMM as other baselines.

In this experiment, we used the original form of the SDGM. To achieve sparse optimization during end-to-end training, we employed an approximated sparse Bayesian learning based on Gaussian dropout proposed by Molchanov et al. (2017). This is because it is difficult to execute the learning algorithm in Section 3.4 with backpropagation due to large computational costs for inverse matrix calculation of the Hessian in (16), which takes $O(N^3)$.

Table 2: Recognition error rates (%) on image classification

|  | MNIST $(D=2)$ | MNIST $(D=10)$ | Fashion MNIST | CIFAR-10 | CIFAR-100 | ImageNet | Normalized mean |
|---|---|---|---|---|---|---|---|
| Softmax | 3.19 | 1.01 | 8.78 | 11.07 | 22.99 | 33.45 | 1.20 |
| Softmax + $L_1$ regularization | 3.70 | 1.58 | 9.20 | 10.30 | 21.56 | 32.43 | 1.29 |
| Large margin softmax | 2.52 | 0.80 | 8.51 | 11.58 | **21.00** | 33.19 | 1.08 |
| Discriminative GMM | 2.43 | **0.72** | 8.30 | **10.05** | 21.93 | 34.75 | 1.05 |
| SDGM | **1.81** | 0.86 | **8.05** | 10.46 | 21.32 | **32.40** | **1.00** |

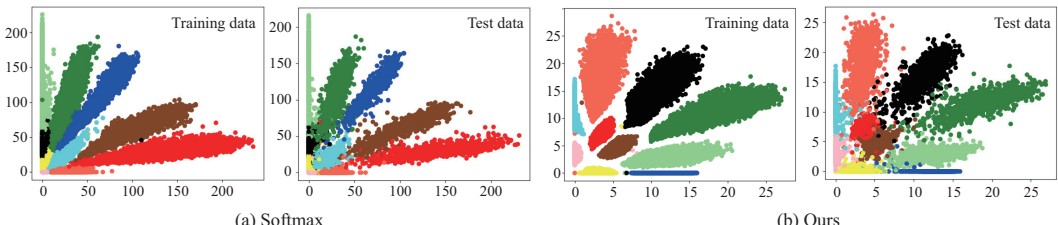

(a) Softmax                   (b) Ours

Figure 5: Visualization of CNN features on MNIST ($D = 2$) after end-to-end learning. In this visualization, five convolutional layers with four max pooling layers between them and a fully connected layer with a two-dimensional output are used. (a) results when a fully connected layer with the softmax function is used as the last layer. (b) when SDGM is used as the last layer instead. The colors red, blue, yellow, pink, green, tomato, saddlebrown, lightgreen, cyan, and black represent classes from 0 to 9, respectively. Note that the ranges of the axis are between (a) and (b).

### 4.2.1 DATASETS AND EXPERIMENTAL SETUPS

We used the following datasets and experimental settings in this experiment.

**MNIST**: This dataset includes 10 classes of handwritten binary digit images of size $28 \times 28$ (LeCun et al. 1998). We used 60,000 images as training data and 10,000 images as testing data. As a feature extractor, we used a simple CNN that consists of five convolutional layers with four max pooling layers between them and a fully connected layer. To visualize the learned CNN features, we first set the output dimension of the fully connected layer of the baseline CNN as two ($D = 2$). Furthermore, we tested by increasing the output dimension of the fully connected layer from two to ten ($D = 10$).

**Fashion-MNIST**: Fashion-MNIST (Xiao et al. 2017) includes 10 classes of binary fashion images with a size of $28 \times 28$. It includes 60,000 images for training data and 10,000 images for testing data. We used the same CNN as in MNIST with 10 as the output dimension.

**CIFAR-10 and CIFAR-100**: CIFAR-10 and CIFAR-100 (Krizhevsky & Hinton 2009) consist of 60,000 $32 \times 32$ color images in 10 classes and 100 classes, respectively. There are 50,000 training images and 10,000 test images for both datasets. For these datasets, we trained DenseNet (Huang et al. 2017) with a depth of 40 and a growth rate of 12 as a baseline CNN.

**ImageNet**: ImageNet classification dataset (Russakovsky et al. 2015) includes 1,000 classes of generic object images with a size of $224 \times 224$. It consists of 1,281,167 training images, 50,000 validation images, and 100,000 test images. For this dataset, we used MobileNet (Howard et al. 2017) as a baseline CNN.

It should be noted that we did not employ additional techniques to increase classification accuracy such as hyperparameter tuning and pre-trained models; therefore, the accuracy of the baseline model did not reach the state-of-the-art. This is because we considered that it is not essential to confirm the effectiveness of the proposed method.

### 4.2.2 RESULTS

Figure 5 shows the two-dimensional feature embeddings on the MNIST dataset. Different feature embeddings were acquired for each method. When softmax was used, the features spread in a fan shape and some parts of the distribution overlapped around the origin. However, when the SDGM was used, the distribution for each class exhibited an ellipse shape and margins appeared between the

class distributions. This is because the SDGM is based on a Gaussian mixture model and functions to push the features into a Gaussian shape.

Table 2 shows the recognition error rates on each dataset. SDGM achieved better performance than softmax. Although sparse learning was ineffective in two out of six comparisons according to the comparison with the discriminative GMM, replacing softmax with SDGM was effective in all the comparisons. As shown in Figure 5, SDGM can create margins between classes by pushing the features into a Gaussian shape. This phenomenon positively affected classification capability. Although large-margin softmax, which has the effect of increasing the margin, and the discriminative GMM, which can represent multimodality, also achieved relatively high accuracy, the SDGM can achieve the same level of accuracy with sparse weights.

## 5 CONCLUSION

In this paper, we proposed a sparse classifier based on a Gaussian mixture model (GMM), which is named sparse discriminative Gaussian mixture (SDGM). In the SDGM, a GMM-based discriminative model was trained by sparse Bayesian learning. This learning algorithm improved the generalization capability by obtaining a sparse solution and automatically determined the number of components by removing redundant components. The SDGM can be embedded into neural networks (NNs) such as convolutional NNs and could be trained in an end-to-end manner.

In the experiments, we demonstrated that the SDGM could reduce the number of weights via sparse Bayesian learning, thereby improving its generalization capability. The comparison using benchmark datasets suggested that SDGM outperforms the conventional kernel Bayesian classifiers. We also demonstrated that SDGM outperformed the fully connected layer with the softmax function when it was used as the last layer of a deep NN.

One of the limitations of this study is that the proposed sparse learning reduces redundant Gaussian components but cannot obtain the optimal number of components, which should be improved in future work. Since the learning of the proposed method can be interpreted as the incorporation of the EM algorithm into the sparse Bayesian learning, we will tackle a theoretical analysis by utilizing the proofs for the EM algorithm (Wu 1983) and the sparse Bayesian learning (Faul & Tipping 2001). Furthermore, we would like to tackle the theoretical analysis of error bounds using the PAC-Bayesian theorem. We will also develop a sparse learning algorithm for a whole deep NN structure including the feature extraction part. This will improve the ability of the CNN for larger data classification. Further applications using the probabilistic property of the proposed model such as semi-supervised learning, uncertainty estimation, and confidence calibration will be considered.

ACKNOWLEDGMENTS

This work was supported in part by JSPS KAKENHI Grant Number JP17K12752 and JST ACT-I Grant Number JPMJPR18UO.

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

SUPPLEMENTARY MATERIALS

## A  RELATIONSHIP BETWEEN THE DISCRIMINATIVE GAUSSIAN AND LOGISTIC REGRESSION

We show that a fully connected layer with the softmax function, or logistic regression, can be regarded as a discriminative model based on a Gaussian distribution by utilizing transformation of the equations. Let us consider a case in which the class-conditional probability $P(\boldsymbol{x}|c)$ is a Gaussian distribution. In this case, we can omit $m$ from the equations (3)–(6).

If all classes share the same covariance matrix and the mixture weight $\pi_{cm}$, the terms $\pi_{cm}$ in (1), $x_1^2, x_1 x_2, \ldots, x_1 x_D, x_2^2, x_2 x_3, \ldots, x_2 x_D, \ldots, x_D^2$ in (2), and $-\frac{1}{2} s_{c11}, \ldots, -\frac{1}{2} s_{cDD}$ in (6) can be canceled; hence the calculation of the posterior probability $P(c|\boldsymbol{x})$ is also simplified as

$$P(c|\boldsymbol{x}) = \frac{\exp(\boldsymbol{w}_c{}^{\mathrm{T}} \boldsymbol{\phi})}{\sum_{c=1}^{C} \exp(\boldsymbol{w}_c{}^{\mathrm{T}} \boldsymbol{\phi})},$$

where

$$\boldsymbol{w}_c = [\log P(c) - \frac{1}{2} \sum_{i=1}^{D} \sum_{j=1}^{D} s_{cij} \mu_{ci} \mu_{cj} + \frac{D}{2} \log 2\pi + \frac{1}{2} \log |\boldsymbol{\Sigma_c}|, \sum_{i=1}^{D} s_{ci1} \mu_{ci}, \cdots, \sum_{i=1}^{D} s_{ciD} \mu_{ci}]^{\mathrm{T}},$$

$$\boldsymbol{\phi} = \left[1, \boldsymbol{x}^{\mathrm{T}}\right]^{\mathrm{T}}.$$

This is equivalent to a fully connected layer with the softmax function, or linear logistic regression.

## B  EVALUATION OF CHARACTERISTICS USING SYNTHETIC DATA

To evaluate the characteristics of the SDGM, we conducted classification experiments using synthetic data. The dataset comprises two classes. The data were sampled from a Gaussian mixture model with eight components for each class. The numbers of training data and test data were 320 and 1,600, respectively. The scatter plot of this dataset is shown in Figure 6.

In the evaluation, we calculated the error rates for the training data and the test data, the number of components after training, the number of nonzero weights after training, and the weight reduction ratio (the ratio of the number of the nonzero weights to the number of initial weights), by varying the number of initial components as $2, 4, 8, \ldots, 20$. We repeated evaluation five times while regenerating the training and test data and calculated the average value for each evaluation criterion. We used the dual form of the SDGM in this experiment.

Figure 6 displays the changes in the learned class boundaries according to the number of initial components. When the number of components is small, such as that shown in Figure 6(a), the decision boundary is simple; therefore, the classification performance is insufficient. However, according to the increase in the number of components, the decision boundary fits the actual class boundaries. It is noteworthy that the SDGM learns the GMM as a discriminative model instead of a generative model; an appropriate decision boundary was obtained even if the number of components for the model is less than the actual number (e.g., 6(c)).

Figure 7 shows the evaluation results of the characteristics. Figures 7(a), (b), (c), and (d) show the recognition error rate, number of components after training, number of nonzero weights after training, and weight reduction ratio, respectively. The horizontal axis shows the number of initial components in all the graphs.

In Figure 7(a), the recognition error rates for the training data and test data are almost the same with the few numbers of components and decrease according to the increase in the number of initial components while it is 2 to 6. This implied that the representation capability was insufficient when the number of components was small, and that the network could not accurately separate the classes. Meanwhile, changes in the training and test error rates were both flat when the number of initial components exceeded eight, even though the test error rates were slightly higher than the training error rate. In general, the training error decreases and the test error increases when the complexity of

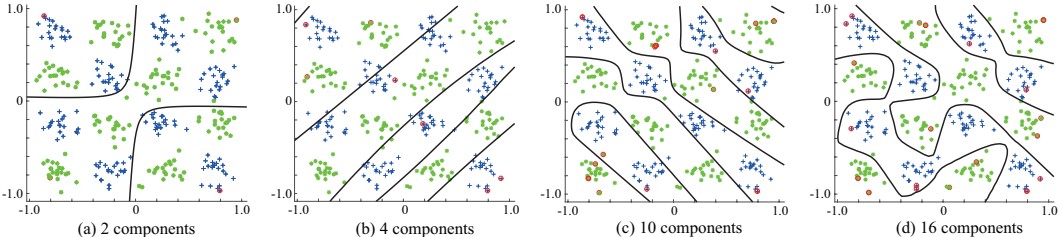

Figure 6: Changes in learned class boundaries according to the number of initial components. The blue and green markers represent the samples from class 1 and class 2, respectively. Samples in red circles represent relevant vectors. The black lines are class boundaries where $P(c \mid \boldsymbol{x}) = 0.5$.

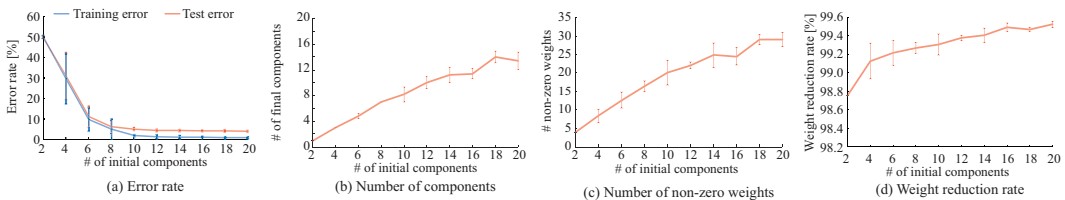

Figure 7: Evaluation results using synthetic data. (a) recognition error rate, (b) the number of components after training, (c) the number of nonzero weights after training, and (d) weight reduction ratio. The error bars indicate the standard deviation for five trials.

the classifier is increased. However, the SDGM suppresses the increase in complexity using sparse Bayesian learning, thereby preventing overfitting.

In Figure 7(b), the number of components after training corresponds to the number of initial components until the number of initial components is eight. When the number of initial components exceeds ten, the number of components after training tends to be reduced. In particular, eight components are reduced when the number of initial components is 20. The results above indicate the SDGM can reduce unnecessary components.

From the results in Figure 7(c), we confirm that the number of nonzero weights after training increases according to the increase in the number of initial components. This implies that the complexity of the trained model depends on the number of initial components, and that the minimum number of components is not always obtained.

Meanwhile, in Figure 7(d), the weight reduction ratio increases according to the increase in the number of initial components. This result suggests that the larger the number of initial weights, the more weights were reduced. Moreover, the weight reduction ratio is greater than 99 % in any case. The results above indicate that the SDGM can prevent overfitting by obtaining high sparsity and can reduce unnecessary components.

## C    DETAILS OF INITIALIZATION

In the experiments during this study, each trainable parameters for the $m$-th component of the $c$-th class were initialized as follows ($H = 1 + D(D+3)/2$, where $D$ is the input dimension, for the original form and $H = N$, where $N$ is the number of the training data, for the kernelized form):

- $\boldsymbol{w}_{cm}$ (for the original form): A zero vector $\boldsymbol{0} \in \mathbb{R}^{H}$.

- $\boldsymbol{\psi}_{cm}$ (for the kernelized form): A zero vector $\boldsymbol{0} \in \mathbb{R}^{H}$.

- $\boldsymbol{\alpha}_{cm}$: An all-ones vector $\boldsymbol{1} \in \mathbb{R}^{H}$.

- $\pi_{cm}$: A scalar $\frac{1}{\sum_{c=1}^{C} M_c}$, where $C$ is the number of classes and $M_c$ is the number of components for the $c$-th class.

- $r_{ncm}$: Initialized based on the results of $k$-means clustering applied to the training data; $r_{ncm} = 1$ if the $n$-th sample belongs to class $c$ and is assigned to the $m$-th component by $k$-means clustering, $r_{ncm} = 0$ otherwise.

