# OpenReview forum: "A Discriminative Gaussian Mixture Model with Sparsity"
_ICLR.cc/2021/Conference — ICLR 2021 Poster_

### Official Review · AnonReviewer1 · 2020-10-16
**A Gaussian mixture model for a better penultimate layer in deep net classifiers**

**Rating:** 8
**Confidence:** 5

**Review:**

The innovative part of the work consists in the trick (8), which allows for a kernel-based generalisation of Gaussian mixture models viewed under a discriminative perspective.

The model is trained via maximum a posteriori estimation; due to the analytically intractable form of the posterior expectation of the log-likelihood, the authors resort to a second-order Taylor approximation around the mode (Laplace approximation). That's an old and not so fine approximation, but easy and widely-known.

The experiments are performed on standard benchmarks, and comparisons are provided against some natural competitors of the method. In essence, the authors replace the penultimate softmax layer of a deep net classifier with their model and train end-to-end.

The experiments yield a marginal improvement over s.o.t.a., as so many papers do nowadays. An aspect that is missing is how they initialised the models. Initialisation may  play a crucial role in such approaches, especially when it comes to ensuring reproducibility of the results.

In summary, a slightly novel paper with some interesting insights and some pretty standard nowadays, yet marginally impressive experimental results.

---

> ### Author Response · Authors · 2020-11-18
> **Response to AnonReviewer1**
>
> **1. The innovative part of the work consists in the trick (8), which allows for a kernel-based generalisation of Gaussian mixture models viewed under a discriminative perspective.**
> **A.** Thank you very much for your positive evaluation and professional comment. The trick (8) is exactly the key to the connection between the discriminative GMM and the Bayesian kernel model, which is one of the contributions of this paper. We are glad that you acknowledged it.
>
> **2. An aspect that is missing is how they initialised the models. Initialisation may play a crucial role in such approaches, especially when it comes to ensuring reproducibility of the results.**
> **A.** Thank you for your suggestion. For reproducibility and learning stability, the weights **$w_{cm}$** and **$\psi_{cm}$**, precision parameter **$\alpha_{cm}$**, and mixture weight $\pi_{cm}$ are initialized with constant values. The variable $r_{ncm}$, which represents the component assignment of each training sample, is initialized based on the results of $k$-means clustering applied to the training data.
> In the revised manuscript, we have added the details of the initialization to Appendix C.
> **[Revision]**
> Appendix C:
> In the experiments during this study, each trainable parameters for the $m$-th component of the $c$-th class were initialized as follows ($H = 1+D(D+3)/2$, where $D$ is the input dimension, for the original form and $H = N$, where $N$ is the number of the training data, for the kernelized form):
> - **$w_{cm}$** (for the original form): A zero vector **$0$** $\in \mathbb{R}^{H}$.
> - **$\psi_{cm}$** (for the kernelized form): A zero vector **$0$** $\in \mathbb{R}^{H}$.
> - **$\alpha_{cm}$**: An all-ones vector **$1$** $\in \mathbb{R}^{H}$.
> - **$\pi_{cm}$**: A scalar $\frac{1}{\sum_{c=1}^C{M_c}}$, where $C$ is the number of classes and $M_c$ is the number of components for the $c$-th class.
> - $r_{ncm}$: Initialized based on the results of $k$-means clustering applied to the training data; $r_{ncm} = 1$ if the $n$-th sample belongs to class $c$ and is assigned to the $m$-th component by $k$-means clustering, $r_{ncm} = 0$ otherwise.

---

> > ### Comment · AnonReviewer1 · 2020-11-18
> > **The rebuttal addresses my concerns.**
> >
> > The rebuttal addresses my concerns.

---

> > > ### Author Response · Authors · 2020-11-25
> > > **Response 2 to AnonReviewer1**
> > >
> > > Thank you very much for your very positive evaluation.

---

### Official Review · AnonReviewer3 · 2020-10-28
**interesting but needs more theory**

**Rating:** 5
**Confidence:** 3

**Review:**

* quality
The paper presents an interesting idea that uses sparse Gaussian mixtures, but it lacks theoretical guarantees. Although the method is Bayesian, can we also give frequentist non-asymptotic bounds?
* clarity
The paper is well written.
* originality
The paper's ideas seem original, but they're very straightforward, making its contributions marginally incremental.
* significance
If the paper had more theoretical guarantees, its results would be more significant. The current version is a bit weak.

---

> ### Author Response · Authors · 2020-11-18
> **Response to AnonReviewer3**
>
> **Although the method is Bayesian, can we also give frequentist non-asymptotic bounds?**
> **A.** Thank you very much for your professional comment. At least for now, we do not think we can give frequentist non-asymptotic bounds because our method is based on Bayesian estimation. Instead, there is a possibility that we can give error bounds for the proposed method using the PAC-Bayesian theorem because generalization error bounds for the Gaussian process classifier were given by Seeger [A4].
> In future work, we would like to tackle the theoretical analysis of error bounds using the PAC-Bayesian theorem. We have modified the manuscript in response to your valuable comment.
> [A4] Seeger, PAC-Bayesian Generalisation Error Bounds for Gaussian Process Classification, Journal of Machine Learning Research, 2002.
> **[Revision]**
> p. 9: Added “Furthermore, we would like to tackle the theoretical analysis of error bounds using the PAC-Bayesian theorem.”

---

### Official Review · AnonReviewer4 · 2020-10-28
**Review of "A Discriminative Gaussian Mixture Model with Sparsity"; borderline accept recommendation.**

**Rating:** 7
**Confidence:** 4

**Review:**

The paper proposes an interesting extension to both discriminative GMMs and CNNs with a softmax outuput. The key innovation is the introduction of the sparsity in discriminative, multimodal settings. This novelty and the clear experiments merit this work to be published at ICLR 21. However, the testing and evaluation could be significantly improved.

There are a few areas where the paper could be improved.
1. The language could use another review. Often, the writing slips into the use of informal language.
2. Appendices were missing in my version; the paper makes an explicit reference to at least appendix A.
3. The connection to Bayesian methods is week.
4. The table describing Algorithm 1 appears abruptly and references formulae that have not been introduced. It is best positioned at the end of Section 3 to improve readability.
5. Most of the Bayesian approaches used in the comparison are dated.
6. Section 4.2 references high computational costs, but it is unclear which steps of the algorithm make this approach computationally prohibitive.
7. It is unclear from Table 1 that SDGM is significantly better than RVM. A deep dive is warranted to better understand their relative performances.
8. It would be better if Section 4.2.1 also include large datasets or more challenging domains. The relative error reduction described in Table 2 on a couple of datasets are marginal.

---

> ### Author Response · Authors · 2020-11-18
> **Response to AnonReviewer4**
>
> Thank you very much for the positive evaluation.
> **1. The language could use another review. Often, the writing slips into the use of informal language.**
> **A.** Thank you for pointing it out. We will use another professional language editing service and reflect the modification by the end of the discussion period.
>
> **2. Appendices were missing in my version; the paper makes an explicit reference to at least appendix A.**
> **A.** We are sorry for making you confused. Appendices were included in supplementary materials. We also added appendices to the main PDF (the official FAQ says either is allowed).
>
> **3. The connection to Bayesian methods is week.**
> **A.** The proposed SDGM is closely related to Bayesian methods. The SDGM is regarded as one of the Bayesian methods because the posterior distribution of weights is estimated during learning instead of directly estimating the weights as points, as with other Bayesian methods. To clarify this, we have updated Section 2.
> **[Revision]**
> p. 3: Added “The SDGM with kernelization is also regarded as a kernel Bayesian method because the posterior distribution of weights is estimated during learning instead of directly estimating the weights as points, as with the GP and MGP.”
>
> **4. The table describing Algorithm 1 appears abruptly and references formulae that have not been introduced. It is best positioned at the end of Section 3 to improve readability.**
> **A.** Thank you for your valuable suggestion. We moved Algorithm 1 to the end of Section 3.
>
> **5. Most of the Bayesian approaches used in the comparison are dated.**
> **A.** Thank you for your insightful comment. As pointed out, the Bayesian methods used in the comparisons (Gaussian process, mixture of Gaussian processes, and relevance vector machine) are a bit old. However, we could not find a method better suited from the perspective of sparsity, multimodality, and Bayesian learning despite our best efforts.
>  In the revised manuscript, we have added the reason why we chose the comparative methods.
> **[Revision]**
> p. 7: Added “For comparison, we used three kernel Bayesian methods: a GP classifier, an MPG classifier (Tresp2001; Luo & Sun 2017), and an RVM (Tipping 2001), which are closely related to the SDGM from the perspective of sparsity, multimodality, and Bayesian learning, as described in Section 2.”
>
> **6. Section 4.2 references high computational costs, but it is unclear which steps of the algorithm make this approach computationally prohibitive.**
> **A.** The most computationally expensive part is inverse matrix calculation of the Hessian (Eq. (16) in the paper), which takes $O(N^3)$, where $N$ is the number of training data. For this problem, we can employ Molchanov's approximation based on dropout [A3], which is almost as computationally inexpensive as softmax training, and we used the approximation in the image classification in Section 4.2. However, the trained SDGM becomes a memory-efficient classifier because it reduces the number of weights by the Sparse Bayesian learning as shown in Table 1 in the paper.
>  In the revised manuscript, we have added some explanation about computational costs.
> [A3] Molchanov et al., Variational dropout sparsifies deep neuralnetworks, in Proc. ICML, 2017.
> **[Revision]**
> p. 7: This is because it is difficult to execute the learning algorithm in Section 3.4 with backpropagation due to large computational costs for inverse matrix calculation of the Hessian in (16), which takes $O(N^3)$.
>
> **7. It is unclear from Table 1 that SDGM is significantly better than RVM. A deep dive is warranted to better understand their relative performances.**
> **A.** We consider that the key is multimodality. For data with a multimodal structure, the decision boundary between classes may have a complex shape at the border of the clusters. As shown in Figure 1 of the paper, the SDGM could more accurately represent the sharp changes in decision boundaries near the border of components compared to the RVM since the SDGM explicitly models multimodality.
>  In the revised manuscript, we have added some explanation to Section 4.1.
> **[Revision]**
> p. 7: Added: Since the SDGM explicitly models multimodality, it could more accurately represent the sharp changes in decision boundaries near the border of components compared to the RVM, as shown in Figure 1.
>
> **8. It would be better if Section 4.2.1 also include large datasets or more challenging domains. The relative error reduction described in Table 2 on a couple of datasets are marginal.**
> **A.** Thank you for your valuable suggestion. As stated in the general response, we will add the results of additional experiments if we can complete the experiments by the end of the rebuttal period.

---

> > ### Comment · AnonReviewer4 · 2020-11-18
> > **Many concerns addressed in the revision**
> >
> > The authors have addressed most of the concerns raised by me: much appreciated. I still think the responses to concerns #3, #5, and #8 are inadequate. I am particularly hopeful of the additional results becoming available: that is my most significant remaining concern.

---

> > > ### Author Response · Authors · 2020-11-25
> > > **Response to AnonReviewer4**
> > >
> > > Thank you for your response. As stated in the general response, we added experiments on CIFAR-100 to Table 2. In response to another reviewer, we will add the results on ImageNet to the camera-ready if the paper is accepted.
> > > [Revision]
> > > p. 8: Added the results on CIFAR-100 to Table 2.
> > >
> > > Table 2. Recognition error rates (%) on image classification
> > >
> > > |                                      | MNIST (D = 2) | MNIST (D = 10) | Fashion-MNIST | CIFAR-10  | CIFAR-100 | Normalized mean |
> > > | ------------------------------------ | ------------- | -------------- | ------------- | --------- | --------- | --------------- |
> > > | CNN + Softmax                        | 9.19          | 1.01           | 8.78          | 11.07     | 22.99     | 1.23            |
> > > | CNN + Softmax + $L_1$ regularization | 3.70          | 1.58           | 9.20          | 10.30     | 21.56     | 1.40            |
> > > | CNN + Large-margin softmax           | 2.52          | 0.80           | 8.51          | 11.58     | **21.00** | 1.09            |
> > > | CNN + Discriminative GMM             | 2.43          | **0.72**       | 8.30          | **10.05** | 21.93     | 1.04            |
> > > | CNN + SDGM                           | **1.81**      | 0.86           | **8.05**      | 10.46     | 21.32     | **1.00**        |

---

### Official Review · AnonReviewer2 · 2020-10-29
**A Discriminative Gaussian Mixture Model with Sparsity**

**Rating:** 6
**Confidence:** 4

**Review:**

The paper proposes a sparse classifier via  discriminative GMM. This model is trained based on sparse Bayesian learning. The sparsity constraint removes redundant Gaussian components which results in  reducing the number of parameters and improving the generalization. This framework can potentially be embedded into the deep models and trained in an end-to-end fashion. The main motivation is that the proposed model (i.e., SDGM,)  can consider multimodal data while conventional softmax classifiers only assume unimodality for each class. Experimental results show the superiority of the SDGM over existing softmax-based discriminative models.

The paper is well-written and easy to follow. And the paper precisely places the proposed method among the related work.  However, there are some concerns:

1- Sparsity constraint is supposed to improve the performance and generalization, but experimental results do not support this. What is the motivation of employing sparsity learning in this framework?

2- Conventional softmax classifiers in deep architecture have already provided promising results on real world datasets such as ImageNet which contains classes with multimodal data. However, in this paper, experiments are performed on the small datasets, so was the proposed method evaluated on ImageNet as well?


3- There are many versions of softmax-classifiers such as “large-margin softmax”, “angular softmax”, and “additive margin softmax” , and  [1 ,2 ] that address conventional softmax-classifiers’ issues. Have you compared your models with them?

[1] Liu W, Wen Y, Yu Z, Yang M. Large-margin softmax loss for convolutional neural networks. InICML 2016 Jun 19 (Vol. 2, No. 3, p. 7).

[2] Weiyang Liu, Yandong Wen, Zhiding Yu, Ming Li, Bhiksha Raj, and Le Song. SphereFace: Deep hypersphere embedding for face recognition. In Proceedings of the IEEE Conference on Computer Vision and Pattern Recognition, pp. 212–220, 2017

[3] Feng Wang, Jian Cheng, Weiyang Liu, and Haijun Liu. Additive margin softmax for face verification. IEEE Signal Processing Letters, 25(7):926–930, 2018.

[4] Kaidi Cao, Colin Wei, Adrien Gaidon, Nikos Arechiga, and Tengyu Ma. Learning imbalanced datasets with label-distribution-aware margin loss. In Advances in Neural Information Processing Systems, pp. 1567–1578, 2019.

[5] Bin Liu, Yue Cao, Yutong Lin, Qi Li, Zheng Zhang, Mingsheng Long, and Han Hu. Negative margin matters: Understanding margin in few-shot classification. arXiv preprint arXiv:2003.12060, 2020.

4-  Finally, I could not find any guarantee for the convergence of the learning algorithm in the paper? What is the time-complexity?  I think training with softmax would be much faster, easier and also provides promising classification results in practice.

---

> ### Author Response · Authors · 2020-11-18
> **Response to AnonReviewer2 (2/2)**
>
> **4. Finally, I could not find any guarantee for the convergence of the learning algorithm in the paper? What is the time-complexity? I think training with softmax would be much faster, easier and also provides promising classification results in practice.**
> **A.** The learning of the proposed method can be interpreted as the incorporation of the EM algorithm into the sparse Bayesian learning. Since the convergence of each algorithm has been proven [A1, A2], it is expected that the learning of the proposed method also converges. In fact, the learning of the proposed method converged in all the experiments in the paper.
> For time complexity, the most computationally expensive part is inverse matrix calculation of the Hessian (Eq. (16) in the paper), which takes $O(N^3)$, where $N$ is the number of training data. For this problem, we can employ Molchanov's approximation based on dropout [A3], which is almost as computationally inexpensive as softmax training, and we used the approximation in the image classification in Section 4.2. However, the trained SDGM becomes a memory-efficient classifier because it reduces the number of weights by the Sparse Bayesian learning as shown in Table 1 in the paper.
> In the revised manuscript, we have added some explanation about learning convergence and time complexity.
> [A1] Faul and Tipping, Analysis of Sparse Bayesian Learning, in Proc. NIPS, 2002.
> [A2] Wu, On the Convergence Properties of the EM Algorithm, Annals of Statistics, 1983.
> [A3] Molchanov et al., Variational dropout sparsifies deep neuralnetworks, in Proc. ICML, 2017.
> **[Revision]**
> p. 7: This is because it is difficult to execute the learning algorithm in Section 3.4 with backpropagation due to large computational costs for inverse matrix calculation of the Hessian in (16), which takes $O(N^3)$.
> p. 9: Added: “Since the learning of the proposed method can be interpreted as the incorporation of the EM algorithm into the sparse Bayesian learning, we will tackle a theoretical analysis by utilizing the proofs for the EM algorithm (Wu 1983) and the sparse Bayesian learning (Faul & Tipping 2001).

---

> > ### Comment · AnonReviewer2 · 2020-11-19
> > **After rebuttal**
> >
> > Thank you authors for the responses. My concern regarding the convergence of the algorithm,  and knowing if the method is useful in terms of time complexity compared to the regular softmax-classifier are addressed. Thus I partially increase my score. Moreover,  it is a common fact that sparsity learning reduces the parameters of the model during training and then it is memory efficient. The efficiency of the methods all depends on how much parameters are reduced and then method needs further emphasize on this. Also the paper deserves a score 7 if the paper adds the results on ImageNet and other softmax-classifier baselines as I mentioned in my review.

---

> > > ### Author Response · Authors · 2020-11-25
> > > **Response to AnonReviewer2**
> > >
> > > Thank you again for your positive evaluation and valuable suggestions. In response to your previous suggestion #3, we added a comparison with large-margin softmax in the image classification experiment. The results showed that the proposed SDGM achieved slightly better performance than large-margin softmax on average even though the SDGM uses sparse weights.
> > > **[Revision]**
> > > p. 8: Added the results of large-margin softmax to Table 2 and related description to Section 4.2.2, respectively.
> > > Table 2. Recognition error rates (%) on image classification
> > >
> > > |                                      | MNIST (D = 2) | MNIST (D = 10) | Fashion-MNIST | CIFAR-10  | CIFAR-100 | Normalized mean |
> > > | ------------------------------------ | ------------- | -------------- | ------------- | --------- | --------- | --------------- |
> > > | CNN + Softmax                        | 9.19          | 1.01           | 8.78          | 11.07     | 22.99     | 1.23            |
> > > | CNN + Softmax + $L_1$ regularization | 3.70          | 1.58           | 9.20          | 10.30     | 21.56     | 1.40            |
> > > | CNN + Large-margin softmax           | 2.52          | 0.80           | 8.51          | 11.58     | **21.00** | 1.09            |
> > > | CNN + Discriminative GMM             | 2.43          | **0.72**       | 8.30          | **10.05** | 21.93     | 1.04            |
> > > | CNN + SDGM                           | **1.81**      | 0.86           | **8.05**      | 10.46     | 21.32     | **1.00**        |
> > >
> > > In response to your previous suggestion #2, we conducted experiments on ImageNet. However, as stated in the general response 2, we could not complete the experiments due to the test submission limit. We would like to add the test results on ImageNet to the camera-ready if the paper is accepted.
> > > In this revision, we have added experiments on CIFAR-100, which consists of 60,000 images in 100 classes, instead of ImageNet. As well as on other datasets, the proposed method achieved relatively good performance on CIFAR-100.
> > > **[Revision]**
> > > p. 8: Added the results on CIFAR-100 to Table 2 and related description to Section 4.2, respectively.
> > >
> > > **The efficiency of the methods all depends on how much parameters are reduced and then method needs further emphasize on this.**
> > > **A.** Thank you for your advice. We emphasized how much parameters are reduced by specifying the ratio of reduced weights.
> > > **[Revision]**
> > > p. 7: Added “the SDGM reduced 90.0--99.5\% of weights from the initial state due to the sparse Bayesian learning”

---

> ### Author Response · Authors · 2020-11-18
> **Response to AnonReviewer2 (1/2)**
>
> **1. Sparsity constraint is supposed to improve the performance and generalization, but experimental results do not support this. What is the motivation of employing sparsity learning in this framework?**
> **A.** Motivations for sparse learning are twofold: memory efficiency and generalization performance. Both are basically supported by the experimental results. Tables R1 and R2 are excerpts from Table 1 in the paper and show the comparison between the proposed SDGM and the mixture of Gaussian processes (MGP), which is equivalently regarded as the SDGM w/o sparse learning, on benchmark datasets. Tables show that the number of weights was remarkably reduced and thus memory efficiency improved, without losing generalization capability on average.
>
> Table R1. Number of weights
>
> |                 | SDGM | MGP    |
> | --------------- | ---- | ------ |
> | Ripley          | 6    | 1250   |
> | Waveform        | 11.0 | 2000   |
> | Banana          | 11.1 | 2000   |
> | Titanic         | 74.5 | 750    |
> | Breast Cancer   | 15.7 | 1000   |
> | Normalized mean | 1.00 | 128.79 |
>
> Table R2. Error rate (%)
>
> |                 | SDGM | MGP  |
> | --------------- | ---- | ---- |
> | Ripley          | 9.1  | 9.3  |
> | Waveform        | 10.1 | 9.6  |
> | Banana          | 10.6 | 10.7 |
> | Titanic         | 22.6 | 22.8 |
> | Breast Cancer   | 29.4 | 30.7 |
> | Normalized mean | 1.00 | 1.01 |
>
> In the revised manuscript, we have clarified the motivations for sparse learning and explained the comparative results with non-sparse classifiers in Section 4.1.
> **[Revision]**
> p. 1: which may lead to overfitting -> which may lead to overfitting and an increase in memory usage
> p. 1: improves the generalization capability -> reduces memory usage without losing generalization capability
> p. 7: Added “Although the SDGM did not necessarily outperform the other methods in all datasets, it achieved the best accuracy on average. In terms of sparsity, the SDGM remarkably reduced the number of non-zero weights compared to non-sparse classifiers (GP and MGP) due to the sparse Bayesian learning, which leads to drastically efficient use of memory.”
>
> **2. Conventional softmax classifiers in deep architecture have already provided promising results on real world datasets such as ImageNet which contains classes with multimodal data. However, in this paper, experiments are performed on the small datasets, so was the proposed method evaluated on ImageNet as well?**
> **A.** Thank you for your valuable suggestion. We have not evaluated the proposed model on ImageNet, and we have started an additional experiment on it. As stated in the general response, we will add the results if we can complete the experiments by the end of the rebuttal period.
>
> **3. There are many versions of softmax-classifiers such as “large-margin softmax”, “angular softmax”, and “additive margin softmax”, and [1, 2] that address conventional softmax-classifiers’ issues. Have you compared your models with them?**
> **A.** We have not compared the proposed method with softmax-classifiers, and we have started a comparative experiment. As stated in the general response, we will add the results if we can complete the experiments by the end of the rebuttal period.

---

### Author Response · Authors · 2020-11-18
**General response**

We thank all the reviewers for their very positive and constructive comments. To our understanding, all the reviewers kindly appreciated our methodology and thus we will be able to have positive evaluations from all the reviewers just by adding more experiments with additional datasets and a comparative method (softmax-classifier) for the further confirmation of practical performance. We have started additional experiments on ImageNet but it will take some time; we will add the results to the revised manuscript if we can complete the experiments by the end of the rebuttal period. Therefore, at present, we first address the remaining concerns. We separately respond to individual reviewers to address their concerns. We also revised the manuscript in response to their suggestions, and text with changes made is colored in red.

---

> ### Comment · Area_Chair1 · 2020-11-18
> **Authors: Thank you for response / Reviewers: Please update**
>
> Thank you, authors, for your responses.
>
> Reviewers, please read the responses and update your reviews by stating that your concerns have been addressed or by providing further rebuttal.

---

> > ### Author Response · Authors · 2020-11-25
> > **General response 2**
> >
> > We thank the reviewers for their insightful comments and positive evaluations. As we stated in the previous rebuttal comments, we conducted additional experiments. In this revision, we added (1) experiments on the CIFAR-100 dataset and (2) large-margin softmax for comparison in Section 4.2.
> >
> > Unfortunately, the experiments on ImageNet (ILSVRC2012) have not been completed in time. This is because of the submission limit of ImageNet. ImageNet does not distribute test labels, and we should upload the predictions of labels by the model to the ImageNet server to obtain test accuracy. The number of submissions is limited to twice per week, and we could not complete the comparison because we have five methods to be compared. Since training of each method has finished, we will add the test results to the camera-ready if the paper is accepted.

---

### Decision · Program_Chairs · 2021-01-07
**Final Decision**

**Decision:**

Accept (Poster)

**Comment:**

This paper proposes replacing the softmax of deep NNs with a kernel-based Gaussian mixture model, to allow for per-class multi-modality.  Results show that the method is competitive with other output modifications such as the large-margin softmax.

The  two primary concerns of the reviewers were the lack of large-scale image classification results and theoretical guarantees.  The authors have added CIFAR-100 results.  Moreover, the authors agree that theoretical results would be nice to have, but such results are non-trivial and likely require a PAC-Bayes treatment.

I find the method to be well-motivated and that the paper demonstrates sufficient experimental rigor.  Given the popularity of the softmax throughout deep learning, this paper will likely be of interest---or at least, be of potential use---to a large part of the ICLR community.  I encourage the authors to add the ImageNet results to the final version.